# Atomic structure of the innexin-6 gap junction channel determined by cryo-EM

Atsunori Oshima[1,2], Kazutoshi Tani[1] & Yoshinori Fujiyoshi[1,2]

Innexins, a large protein family comprising invertebrate gap junction channels, play an essential role in nervous system development and electrical synapse formation. Here we report the cryo-electron microscopy structures of *Caenorhabditis elegans* innexin-6 (INX-6) gap junction channels at atomic resolution. We find that the arrangements of the transmembrane helices and extracellular loops of the INX-6 monomeric structure are highly similar to those of connexin-26 (Cx26), despite the lack of significant sequence similarity. The INX-6 gap junction channel comprises hexadecameric subunits but reveals the N-terminal pore funnel, consistent with Cx26. The helix-rich cytoplasmic loop and C-terminus are intercalated one-by-one through an octameric hemichannel, forming a dome-like entrance that interacts with N-terminal loops in the pore. These observations suggest that the INX-6 cytoplasmic domains are cooperatively associated with the N-terminal funnel conformation, and an essential linkage of the N-terminal with channel activity is presumably preserved across gap junction families.

[1] Cellular and Structural Physiology Institute (CeSPI), Nagoya University, Furo-cho, Chikusa-ku, Nagoya 464-8601, Japan. [2] Department of Basic Medicinal Sciences, Graduate School of Pharmaceutical Sciences, Nagoya University, Furo-cho, Chikusa-ku, Nagoya 464-8601, Japan. Correspondence and requests for materials should be addressed to A.O. (email: atsu@cespi.nagoya-u.ac.jp).

Cell-cell communication through gap junction channels is indispensable for maintaining homeostasis in most multi-cellular living beings. Under electron microscopy at low resolution, the morphologies of gap junction channels from native tissues of vertebrates and invertebrates appear analogous[1]. It is now known, however, that two genetically distant genes code gap junction proteins. Connexins are well investigated because they exist in chordates, including vertebrates, and are associated with human diseases such as hearing loss, cataracts, skin and heart diseases, and oculodentodigital dysplasia[2,3]. Invertebrates possess a different gap junction protein family, termed innexins[4], that is essential for eating and developmental functions in *Caenorhabditis elegans*[5,6] and for electrical synapses in *Drosophila*[7]. Whether connexins and innexins share a common ancestor or arose independently by convergent evolution remains to be determined[8].

Both families contain four transmembrane helices (TM1 to TM4) and two extracellular loops (E1 and E2). X-ray crystal structures of gap junction channels are available only for connexin-26 (Cx26)[9,10]. These structures clarified the assignment of the transmembrane helices in a dodecamer. The N-terminal domains (NTs) of Cx26 form a funnel conformation in a pore cavity, and are thought to be essential for the passage of permeates[9]. The recently determined structure of Cx26 suggests $Ca^{2+}$ binding sites[10]. The detailed features of the cytoplasmic loop (CL) and C-terminal domain (CT) of gap junction channels have not been visualized in any crystallographic studies on connexins[9–14]. We recently revealed that the *C. elegans* innexin-6 (INX-6) gap junction channel structure is a 16-subunit oligomer, that is, a hexadecamer, with larger dimensions than those of connexins[15]. The lack of high-resolution structures of innexins, however, has made it difficult to compare the details of the structural arrangements of innexins and connexins.

A number of membrane protein structures have recently been determined by single-particle analysis cryo-electron microscopy (cryo-EM) at atomic or near atomic resolutions[16–18]. While it remains challenging to prepare perfectly thin ice layers of membrane protein particles originally purified in detergent solution, successful strategies have been developed in which surrounding detergent micelles are exchanged with amphipols[16] or nanodiscs[19], or the detergent-solubilized proteins are used directly for cryo grid preparation[20]. An alternative method, named GraDeR, was recently established, in which free detergent micelles are removed from solubilized membrane complexes by glycerol gradient centrifugation[21]. Here we report the atomic resolution structures of INX-6 determined by cryo-EM in combination with GraDeR.

## Results

**Structure determination of the INX-6 channels**. INX-6 wild-type (WT) channels were purified in 0.1% octyl glucose neopentyl glycol solution[22]. We first prepared cryo grids with the gel filtration eluate, but the cryo images did not exhibit Thon rings at high resolution (Supplementary Fig. 1a). Subsequently, we used GraDeR[21] to exclude detergent micelles as much as possible. After confirming that most of the micelles were removed (Supplementary Fig. 1b), we obtained cryo images in which the particles were recorded with better contrast and the Thon rings were visible at ~3 Å resolution (Supplementary Fig. 1c). In this study, the frequency of ideally thin ice layers containing INX-6 particles was increased by GraDeR about once during a couple of plunge freezing trials, whereas we were unable to obtain such optimal specimens for high-resolution imaging using other methods.

We first attempted to reconstruct a hemichannel of INX-6 as semi-auto particle picking worked very well (Supplementary Fig. 2a). The two-dimensional class averages exhibited stripes, indicating transmembrane helices with helical features (Fig. 1a). A three-dimensional map was reconstructed from 74,398 particles with C8 symmetry, and the final resolution after refinement was 3.3 Å (Fig. 1b, Supplementary Figs 2b,d,3a,b,4a,b and Supplementary Table 1), which was sufficient for *de novo* model building (Fig. 1c,d). The map of the INX-6 hemichannel includes part of the densities of the extracellular domains of an opposed hemichannel (Fig. 1b). To construct a full gap junction channel of INX-6, particles showing the properties of docked junction channels were picked from the same set of cryo images (Supplementary Fig. 2a). The density map of the INX-6 gap junction channel was reconstructed at 3.6 Å resolution from 35,608 particles with D8 symmetry (Fig. 1e, Supplementary Figs 2c,3a,b, and Supplementary Table 1). A model of the INX-6 gap junction channel was refined after fitting the atomic model of the INX-6 hemichannel into the density map of the INX-6 gap junction channel (Fig. 1e). The Met1 to Gly6 residues at the N-terminal end, Ile52 and Gly53 following TM1, and Glu370 to the C-terminal end were not assigned due to the disorder of the structure. The processing schemes we used for the hemi-channel and gap junction channel are summarized in Supplementary Fig. 5.

**Monomeric structure of INX-6**. Although it is considered that there is no significant sequence similarity between connexin and innexin[8], several aspects of the INX-6 monomeric structure were similar to those of Cx26. Specifically, the N-terminus has a short helix (NTH) facing the pore. The assigned order of transmembrane helices is the same, and the innermost helix is TM1. TM2 is kinked by a proline. E1 contains a small α-helix (E1H), and anti-parallel β-sheets are formed in E2 (Figs 2a,b and 3a). Two disulfide bonds between E1 and E2 are generated by the pairs of Cys58-Cys265 and Cys76-Cys248 (Fig. 3b, asterisks), which are strongly conserved cysteines in the innexin family (Supplementary Fig. 6, asterisks). This distribution is similar to two of the three disulfide bonds found in Cx26 (ref. 9) (Fig. 3b). E1 of INX-6 has two lobes clamping the E2 anti-parallel β-sheets from both sides (Figs 2a and 3b). The longer E2 residues provide an additional short helix (E2H). The visible CL and CT are unique to INX-6. Two helices (CLH1 and CLH2) in CL and four helices (CTH1 to CTH4) in CT are arranged in a helix-turn-helix motif, aligning vertically to form a cytoplasmic core (Fig. 2a). The connecting points of the cytoplasmic core to the transmembrane helices are kinked like the neck of a sea horse (Fig. 2a, blue arrowhead). The C-terminal loop flanked by CTH3 and CTH4 is within interacting distance to the N-terminal loop (Fig. 2a, red arrowhead). These observations suggest the possibility that conformational changes in the cytoplasmic core are conveyed to the N-terminus (see discussion below).

**Hexadecameric gap junction structure of INX-6**. The hexa-decameric structure of the INX-6 gap junction channel appears similar to the Cx26 structure[9], a tsuzumi-shape (Fig. 4a and Supplementary Fig. 7a). The channel height is estimated to be about 200 Å, which is smaller than that observed in INX-6ΔN[15]. Approximately 20 residues at the C-terminal end are missing and may possibly account for the cytoplasmic bobble found in INX-6δN. The INX-6 structure is divided into three parts: the cytoplasmic dome, the transmembrane region including a pore funnel and the extracellular gap region (Fig. 4a,b). In the cytoplasmic dome, the CLs and CTs are intercalated one by one through the eight subunits, creating a dome-shaped pore entrance

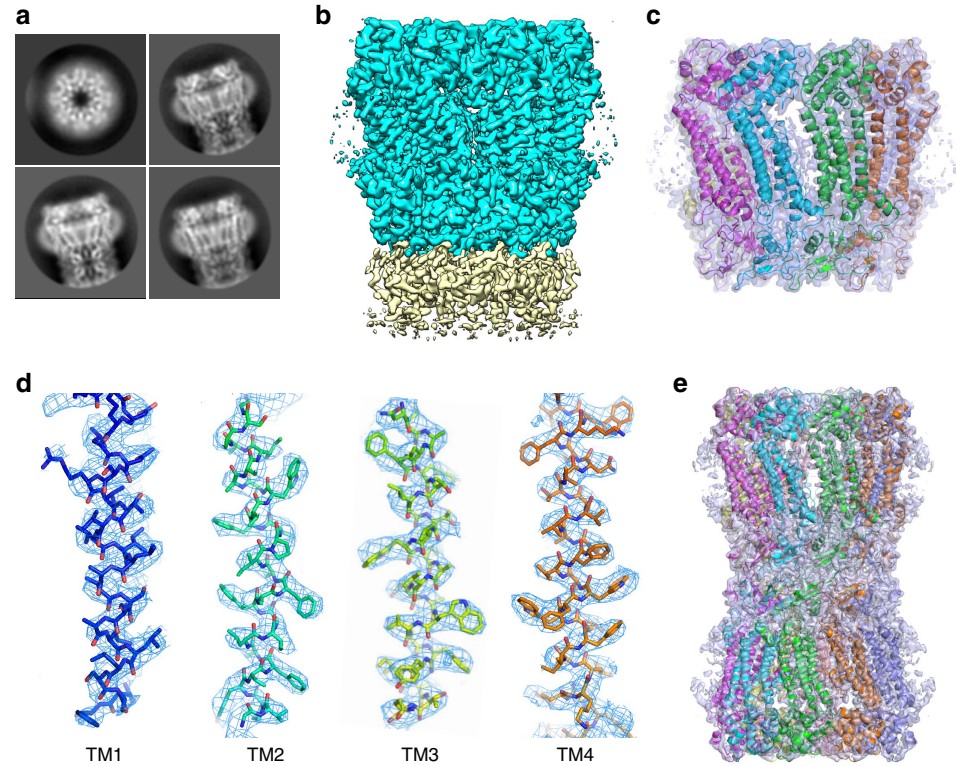

**Figure 1 | Density maps of INX-6 determined by cryo-EM.** (**a**) Representative two-dimensional class averages of INX-6 in top and side views. (**b**) Three-dimensional structure of INX-6 hemichannel (cyan) in a side view. Densities in yellow represent a part of the opposed hemichannel. (**c**) Ribbon style model of INX-6 superimposed on the density map of a hemichannel. (**d**) Density maps of transmembrane helices. Stick style atomic models of TM1-TM4 fit to the cryo-EM density maps (blue mesh). The density map is contoured at 2.0σ. (**e**) Ribbon style model of INX-6 fit on the density map of a gap junction channel.

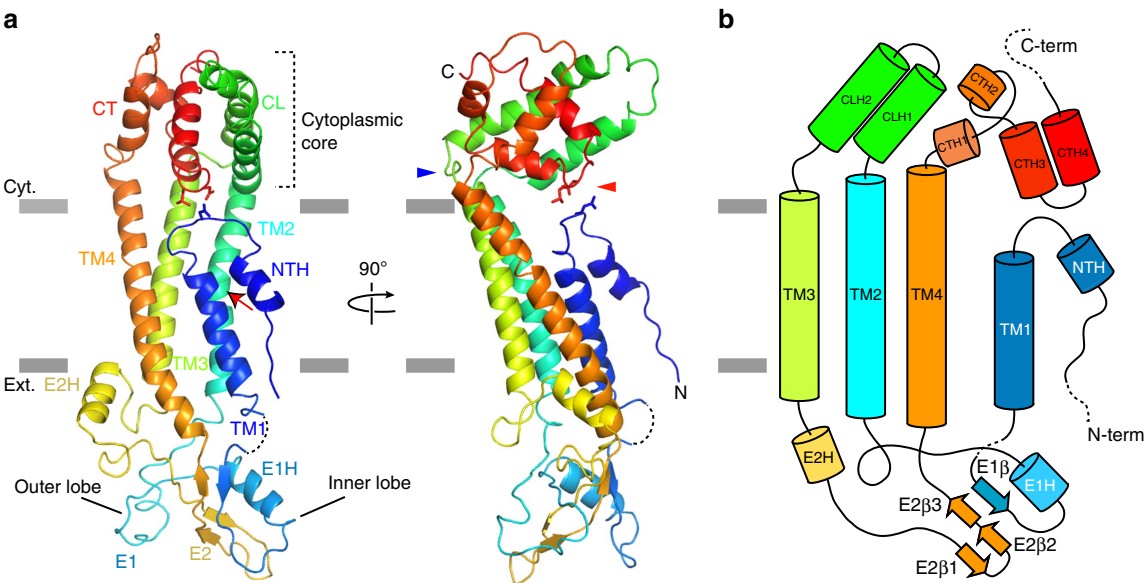

**Figure 2 | Monomeric structure of INX-6.** (**a**) Ribbon representation of the INX-6 monomeric structure. TM2 is kinked at P122 (red arrow). The cytoplasmic core is connected to TM2, TM3 and TM4 with kinked loops (blue arrowhead). The N-terminal loop interacts with CT (red arrowhead) via D25, L347 and N348 depicted in stick style. Grey lines indicate the estimated lipid bilayer borders. (**b**) Schematic representation of secondary structure of INX-6. Dashed lines indicate unassigned residues due to disorder.

with a 30-Å diameter (Fig. 4b—cytoplasmic). The inter-subunit interfaces are reinforced by the polar interactions of Arg168-Asp153, Asp175-Tyr356, Arg178-Thr318, Lys182-Ser317 and Lys183-Asp351 (Fig. 4c), resulting in a roof made of consecutively placed cytoplasmic cores. The spacing between adjacent transmembrane helix bundles is rather curious (Fig. 4d and

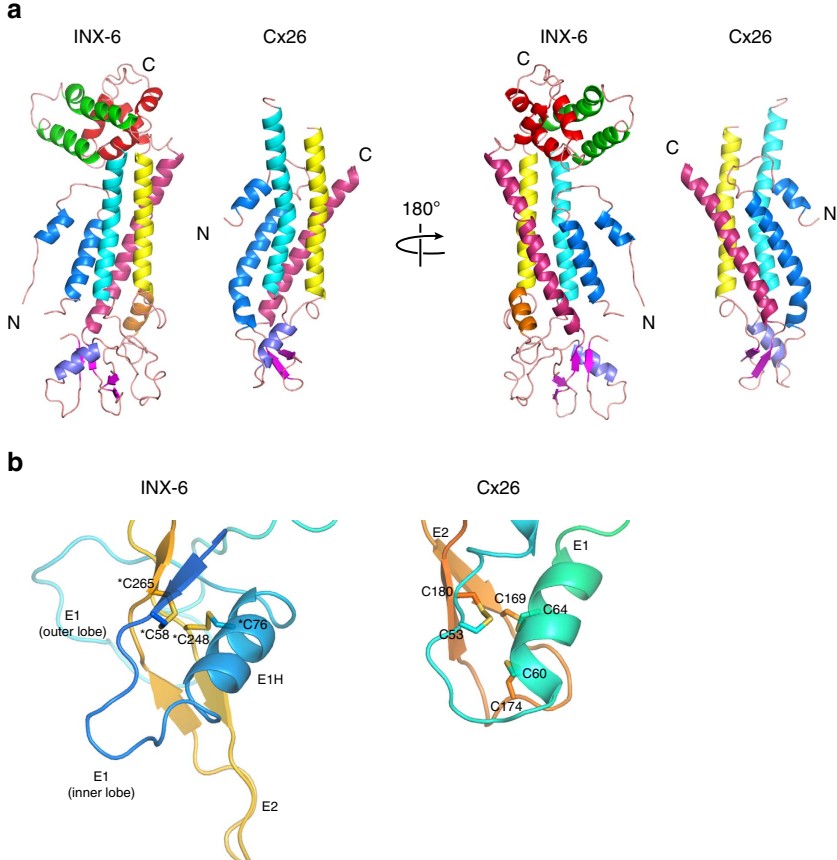

**Figure 3 | Similarity between the INX-6 and Cx26 monomers. (a)** The monomeric structures of INX-6 and Cx26 (pdb code: 2zw3)[9] are presented side by side. Each transmembrane helix is coloured as follows: TM1 in blue, TM2 in cyan, TM3 in yellow and TM4 in pink. β-sheets are coloured magenta. The N- and C- termini are indicated by 'N' and 'C', respectively. The two structures were compared using a Dali server[48], and the Z score was 7.9, suggesting that the protein-folding of the two structures is highly correlated. **(b)** Distributions of disulfide bonds in the extracellular loops of INX-6 and Cx26 when their transmembrane helices are aligned. (left) E1 and E2 of INX-6 are linked by two disulfide bonds of C58-C265 and C76-C248 (asterisks). (right) Three disulfide bonds between E1 and E2 of Cx26 (ref. 9).

Supplementary Figs 4a,b,7b), and suggests flexibility of the helices, as implied previously[15,23]. We found additional densities inserted into the inter-subunit space (Fig. 4d and Supplementary Figs 4b,7b). These densities could be assigned to lipids carried over during purification, which may serve to stabilize the transmembrane bundles and may also contribute to the conformation of the N-terminal funnel due to the proximity. The extracellular domains contain more random coil structures. The pore is restricted, however, by E1Hs in a radial fashion surrounded by anti-parallel β-sheets (Fig. 4b—extracellular). These E1 loops form one of the constrictions in the pore pathway with a 19-Å diameter. Both E1 and E2 contribute to the junctional interfaces. The extracellular docking surface of a hemichannel provides cavities to accommodate the E2 β-hairpins from the opposing hemichannel (Fig. 4e), which is reminiscent of a model previously proposed for connexin[24]. The inner and outer lobes of E1 straddle the E2 β-hairpin stacks whereby the E2 β-hairpins serve as a wall to prevent from leakage of permeates to the extracellular space (Fig. 4e and Supplementary Fig. 7c).

## Discussion

The pore pathway has a high positive potential and is constricted at two sites for each hemichannel (four per junction channel; Supplementary Fig. 8). The narrowest constrictions are provided by the N-termini making a funnel configuration with an 18-Å diameter (Fig. 4b—membrane and Fig. 5a), and others are given

by E1Hs (Fig. 4b—extracellular and Fig. 5a). Previous reports suggest that invertebrate gap junctions possess a larger pore size than vertebrate gap junctions[25–27]. INX-6 channels can also transfer large molecules, such as dextran conjugated to a fluorescent molecule, with a molecular mass over 3000 Da (ref. 22). Although it remains inconclusive whether our structure represents an open or closed conformation because the six N-terminal residues are missing, the diameter at the narrowest constriction ($\sim 18$ Å) will allow for the passage of any hydrated ions and even a single helix peptide ($12 \sim 13$ Å). Although a positively charged environment at the Cx26 channel entrance may possibly contribute to the charge selectivity of permeates[9], the positive pore pathway of INX-6 (Supplementary Fig. 8a) would be less related to the selectivity because of large diameter. Given that the overall structure of INX-6 with an N-terminal funnel and NTHs in a pore cavity is very similar to that of Cx26 (ref. 9), the role that the funnel formation of NTs plays in channel activity is likely to be shared between the two families, despite their different oligomeric numbers and lack of significant sequence similarity. Because pannexins have high homology with innexins in terms of the N-terminal halves[28], the N-terminal of pannexins may also contribute to the pore cavity.

The N-terminal loops are harnessed by the cytoplasmic dome, and the disordered Ile52 and Gly53 cause the flexibility of TM1 in the pore cavity (Fig. 5a). The local resolution map of transmembrane domains of INX-6 indicates that the main chains of NTH and the extracellular side of TM1 are very flexible and

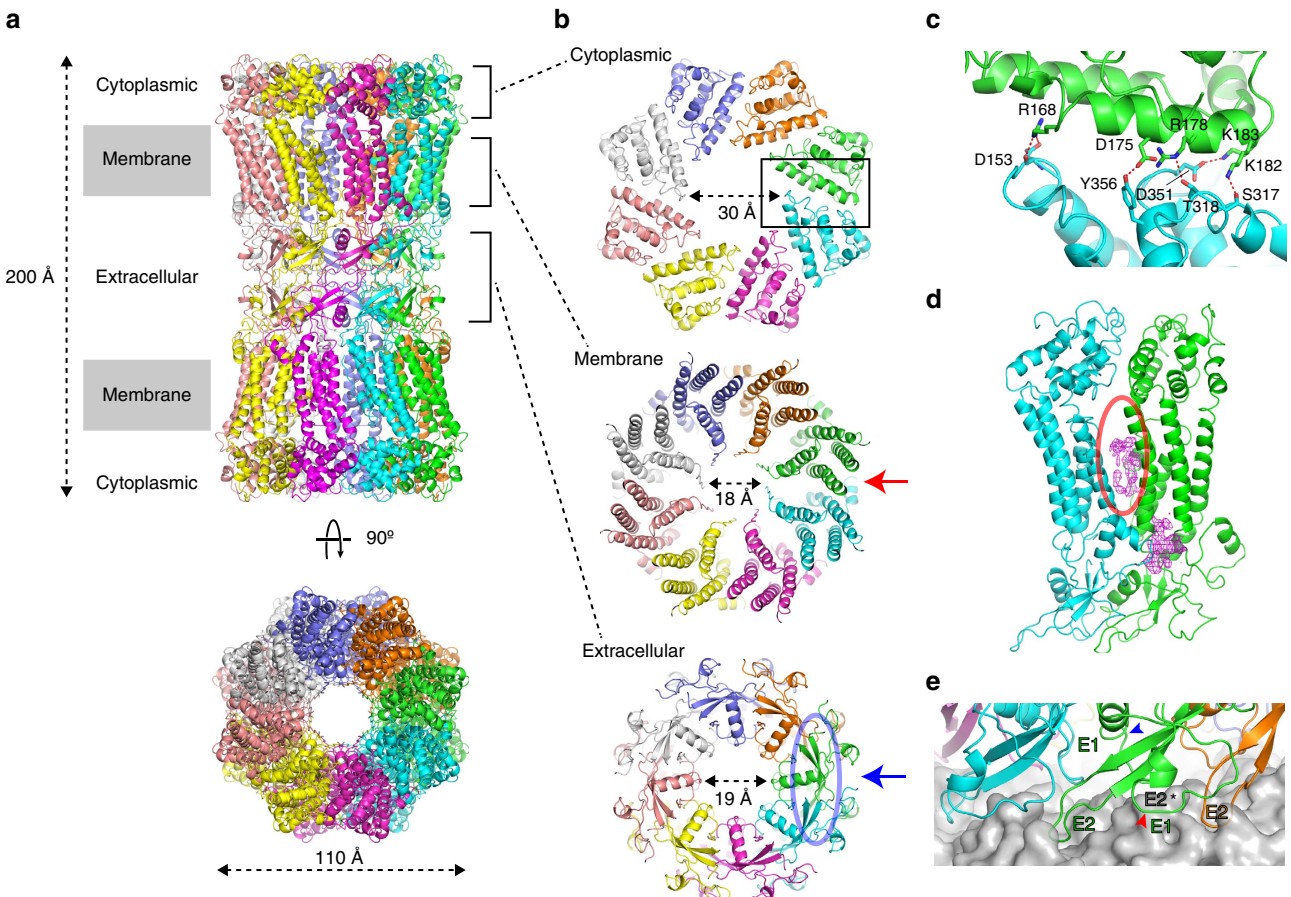

**Figure 4 | INX-6 gap junction structure and domain interface interactions.** (**a**) Ribbon model of INX-6 gap junction channel in side view (top) and top view (bottom). (**b**) Three segments of INX-6. Cytoplasmic dome (cytoplasmic), transmembrane region (membrane) and extracellular gap region (extracellular) correspond to the brackets in **a**. (**c**) Magnified view of the inter-subunit interactions in the cytoplasmic dome indicated by the rectangular box in **b**-cytoplasmic. (**d**) A red circle indicates spacing between the adjacent helix bundles. Unassigned densities (magenta) are located between the two subunits. View orientation is indicated by red arrow in **b**-membrane. (**e**) Extracellular junction interface corresponding to the blue circle in **b**-extracellular where view orientation is indicated by blue arrow. The opposed E2 β-hairpin (asterisk) is fit to a cavity formed by the inner (blue arrowhead) and outer (red arrowhead) lobes of E1 and the two adjacent E2 β-hairpins (green and orange). The molecular surface of an opposed hemichannel is coloured in grey.

that the residues around the interactions between the N-terminal and C-terminal loops exhibit low resolution (Supplementary Fig. 4d). These observations suggest that the N-terminal funnel functions in association with the cytoplasmic dome for the channel activity, and possibly involving TM1 (Fig. 5b). It is known that invertebrate gap junction channels are sensitive to cytoplasmic pH and calcium[29,30]. However, the sensing mechanisms have not been clarified yet. So far no high-resolution structure of connexin demonstrating the CL and CT domains has been reported. For connexin, CL and CT are involved in chemical closure in response to cytoplasmic pH and aminosulfonate[31]. The CL of Cx43 generates two short helical stretches connected by a random coil upon acidification and a direct interaction between the CT and the second half of the CL of Cx43 has been demonstrated[32]. It has also been reported that tethering of the CT of Cx43 to TM4 induces the CT domain to adopt helical structure[33], which suggests the potential of similar cytoplasmic arrangements between Cx43 and INX-6. While other connexins may take different arrangements of the cytoplasmic domains from INX-6 due to a variety of C-terminal lengths, our INX-6 structure demonstrates insights into the cooperation between the cytoplasmic complex and N-terminal funnel of gap junction channels. Further studies will clarify how innexin and

connexin are functionally associated, and why connexins were selected for the chordate gap junction channels in evolutionary history.

## Methods

**Preparation of INX-6 for cryo-EM using GraDeR.** Wild-type INX-6 with GFP plus an 8 × -histidine tag at the C-terminus was isolated and purified from Sf9 cells as described previously[15,22]. In summary, Sf9 cells cultured at 27 °C were infected with the recombinant virus and harvested after infection for 30 ∼ 40 h. The following processes were all performed at 4 °C. The harvested cells were sonicated for 1.5 min in buffer containing 10 mM Tris (pH 7.5), 150 mM NaCl and 1 mM phenylmethylsulfonyl fluoride, and centrifuged at 22,100g for 25 min. The isolated membranes were solubilized in buffer containing 10 mM Tris (pH 7.5), 150 mM NaCl and 2% n-dodecyl-β-D-maltopyranoside (Anatrace) for 30 min. The mixture was centrifuged at 13,000g for 10 min, and the supernatant was bound to Ni-nitrilotriacetic acid agarose (Qiagen). The protein bound resins were washed with buffer containing 10 mM Tris (pH 7.5), 150 mM NaCl, 10 mM L-histidine and 0.1% digitonin (Wako), and eluted with 300 mM L-histidine.

After digesting the C-terminal tag followed by gel filtration with buffer (10 mM Tris (pH 7.5), 500 mM NaCl and 0.1% octyl glucose neopentyl glycol; Anatrace), the fractions containing the INX-6 channels were subjected to GraDeR[21] with minor modifications. LMNG (lauryl maltose neopentyl glycol, Anatrace) was added to the gel filtration fraction of INX-6 at a final concentration of 0.02%. Two different density gradient buffers were prepared. Buffer A for the top layer contained 10 mM Tris (pH 7.5), 500 mM NaCl, 5% glycerol and 0.003% LMNG. Buffer B for the bottom layer was LMNG-free, and contained 10 mM Tris (pH 7.5), 500 mM NaCl and 25% glycerol. Buffers A and B were stacked in centrifuge tubes

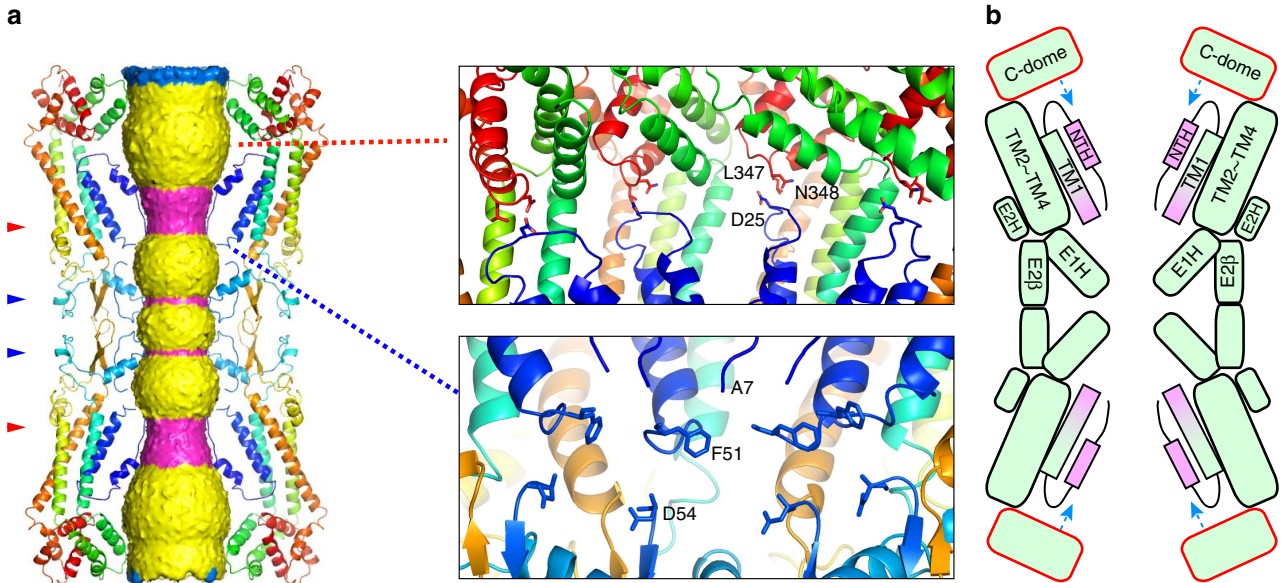

**Figure 5 | Pore pathway and regulation by the cytoplasmic dome of the INX-6 gap junction channel.** (**a**) Surface representation of the pore pathway of the INX-6 gap junction channel. The surface is coloured as follows: magenta, diameter smaller than 19 Å; blue, diameter larger than 36 Å and yellow, diameter between the magenta and blue. Arrowheads indicate the narrowest constrictions (red) and the secondary constrictions (blue). Only four subunits are shown for clarification. (right top) The N-terminal loops are harnessed by the cytoplasmic dome via D25, L347 and N348. (right bottom) TM1 is isolated from E1 due to disordered I52 and G53. (**b**) Schematic representation of the pore funnel regulation model mediated by the cytoplasmic dome (C-dome). The cytoplasmic dome affects the conformation of a pore funnel through the interaction with the loop between NTH and TM1 (blue arrows). Flexible parts of TM1s and NTHs are coloured in pink.

for a Beckman SW41 at a total volume of ∼12 ml. The gradient was generated with a Gradient Master 108 (Biocomp Instruments) using the program for a long cap and glycerol 5%/25% (w/w), with the following parameters: 45 s (time), 86° (angle) and 25 rpm (speed). The INX-6 protein solution (100 ∼ 500 μl in 0.02% LMNG) was carefully loaded onto the top of the gradient buffer without disturbing the gradient solution. Centrifugation was carried out at 35,000 rpm (209,678g) for 12 h at 4 °C. The centrifuged solution was fractionated from bottom to top using a peristaltic pump (Perista Pump, ATTO), and the fractions were subjected to fluorescence-detection size-exclusion chromatography[34] to determine the sedimented position. To remove the glycerol, the fractions containing INX-6 were assembled in a Spectra/pore 7 dialysis tube (molecular weight cutoff: 25,000, Spectrum Laboratories), and dialysed against 1 l of dialysis buffer (10 mM Tris (pH 7.5) and 500 mM NaCl) for 1 h at 4 °C. The dialysed sample was negatively stained with 2% uranyl acetate and observed on a JEM-1010 (JEOL) operated at 100 kV. The INX-6 channels were concentrated to 2 ∼ 3 mg ml$^{-1}$ using a concentrating device with a molecular weight cutoff of 30,000.

**Cryo-EM data acquisition and image processing.** For cryo-EM, pre-irradiated 200 mesh Quantifoil R2/2 molybdenum grids were glow-discharged and 1 ∼ 2 μl of INX-6 channels was placed onto the grid. Excess solution was blotted with a Vitrobot Mark IV (FEI) followed by plunge freezing into liquid ethane. The blotting parameters were as follows: blotting force 1 ∼ 3, blotting time 15 ∼ 20 s and 100% humidity at 22 °C.

Data collection was performed on a JEM-3000SFF (JEOL) electron microscope at 300 kV with a field emission gun and a magnification of × 40,600. The specimen stage temperature was maintained at approximately 80-100 K. The images were recorded on a K2 summit direct electron detector camera (Gatan) operated in a super-resolution mode (7,676 × 7,420 pixels) with a pixel size of 0.616 Å at the specimen level. The dose rate was limited to 10.8 e$^-$ per physical pixel per second, corresponding to 7.1 e$^-$ per Å$^2$ at the specimen. The recording time was 7.2 s after a 0.7-s pre-dose delay, resulting in a total exposure time of 7.9 s and an accumulated dose of 56 e$^-$ per Å$^2$. Each image includes 24 fractioned frames and one frame corresponds to 0.3-s exposure. The defocus ranged from − 0.8 to − 2.4 μm.

For image processing, the dose fractioned image stacks were binned 2 × 2 by Fourier cropping, resulting in 3,838 × 3,710 pixels with a pixel size of 1.232 Å. The stacked frames were subjected to motion correction with MotionCorr[35]. CTF determination was performed using CTFFIND3 (ref. 36). EMAN2 (ref. 37) was used for particle picking and making the initial three-dimensional model. For the INX-6 hemichannel, 1,053 images were processed. In all, 341,585 particles were selected automatically on full-dose (56 e$^-$ per Å$^2$) images and extracted from first 14 frame-projected images (35 e$^-$ per Å$^2$) with a box size of 160 × 160 pixels. The particle set along with an initial three-dimensional model were passed to

RELION[38] to calculate the three-dimensional classification, refinement and post-processing. Three-dimensional classification with C1 symmetry produced five classes. 'Class 1' which showed the finest features of a docked hemichannel structure derived from 74,398 particles and 'class 3' which contained 77,697 particles showing the features of an undocked hemichannel were independently subjected to the next iterative refinement with C8 symmetry. The three-dimensional refinement of 'class 3' reached 7.3 Å resolution with the angular distribution biased toward top views (Supplementary Fig. 2d). The subsequent two-dimensional classification allowed us to select only side view particles from 'class 3'. However, the selected 6,401 particles produced a 6.0 Å resolution map (Supplementary Fig. 2d), which was not included in this work due to the low resolution. Once the refinement using the 'class 1' particles was converged, the particle set was replaced with the subset of the first eight frames (21 e$^-$ per Å$^2$) and refinement was restarted again until convergence. These data were used for post-processing on RELION to obtain the final three-dimensional reconstruction. For the INX-6 gap junction channel, we used 717 images that were part of the same data set for hemichannel analysis. Because auto particle picking does not work well for gap junction channels, manual picking was mainly used in combination with semi-auto picking in a compensative manner. 35,608 INX-6 gap junction channel particles were extracted from images of the 35 e$^-$ per Å$^2$ dose with a box size of 200 × 200 pixels, and an initial three-dimensional model was obtained on EMAN2. In the RELION process, all extracted particles were subjected to iterative refinement with D8 symmetry. In post-processing, soft mask and temperature factor sharpening were applied to both hemi- and gap junction channel maps. Fourier shell correlation and local resolution were calculated on RELION and RESMAP[39], respectively, and the final resolution of INX-6 after masking the map was 3.3 Å for the hemichannel and 3.6 Å for the gap junction channel. Resolutions were assessed by the gold-standard Fourier shell correlation using the criterion of 0.143 (ref. 40).

**Model building of the INX-6 channels.** An initial model of the INX-6 hemichannel was manually built on COOT[41], and refined on PHENIX[42] using real-space refinement. The COOT/PHENIX refinement was iterated for several cycles. Secondary structure and non-crystallographic symmetry restraints were applied throughout the refinement. The initial atomic model of the INX-6 gap junction channel was obtained by fitting the model of the hemichannel to the density map of the INX-6 gap junction channel at 3.6 Å resolution using 'fit-in-map' in UCSF Chimera[43], followed by rigid body refinement with PHENIX. The COOT/PHENIX refinement was performed to build the final model of the gap junction channel. The quality of each model was assessed by MolProbity[44]. The atomic models of the hemichannel and gap junction channels of INX-6 were substantially the same except for the side-chain rotamers of several residues, specifically in the E1 outer lobe. Because the current resolutions were not sufficient to discuss these

differences and the angular distribution of INX-6 gap junction channels is highly biased toward side views (Supplementary Fig. 3a), the hemichannel structure, except for the junction structure and extracellular interactions, was the main focus of this study. The N-terminal 1–6 residues, I52, G53 and C-terminal 370-389 residues were not assigned due to absent densities. The membrane width was estimated by Discovery Studio (BIOVIA). The surface potential map was generated by APBS[45]. Pore sizes were calculated by HOLE[46]. The sequences in the innexin family were aligned with ClustalW[47]. Figures were generated using the Pymol Molecular Graphic System (Schrödinger) or UCSF Chimera.

**Data availability**. Cryo-EM density maps for INX-6 hemichannel and gap junction channel have been deposited in the EMDataBank (http://www.emdata-bank.org/) under accession codes EMD-9570 and EMD-9571. Atomic coordinates have been deposited in the PDB under accession numbers 5H1Q and 5H1R. The data that support the findings of this study are available from the corresponding author on request.

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

## Acknowledgements

We are grateful to K. Kobayashi (JEOL) for maintaining the electron microscopes in excellent condition. This work was supported by Grants-in-Aid for Scientific Research (S), the Japan New Energy and Industrial Technology Development Organization (NEDO), the Japan Agency for Medical Research and Development (AMED) (Y.F.); Grants-in-Aid for Scientific Research (C) (K.T.); Grants-in-Aid for Young Scientists (B), Grants-in-Aid for Scientific Research (C) and the Platform for Drug Design, Discovery, and Development from MEXT, Japan (A.O.).

## Author contributions

A.O. purified proteins and collected cryo-EM data; A.O. and K.T. performed image processing; K.T. performed model building and refinement; A.O. K.T. and Y.F. designed the research and wrote the paper.

## Additional information

tracer molecules in the 300 to 800 Dalton range. *J. Membr. Biol.* **50,** 65–100 (1979).

**Competing financial interests:** The authors declare no competing financial interests.

