## [Peer Review File · Nature Communications]

Reviewers' comments:

Reviewer #1 (Remarks to the Author):

Gap junction mediates cell-cell communication by letting relatively large substrates pass through from cytosol of one cell to another. It plays critical role in maintaining cellular homeostasis in multicellular organisms. The gap junction in vertebrate is named connexin and in invertebrate is named innexin. The two shares no apparent sequence homology, but its overall structural architectures were predicted to be somewhat similar. The atomic structure of connexin-26 was determined by X-ray crystallography a few years ago. However, no structure information of innexin was available, not even its oligomeric state. This manuscript by Oshima et al described the atomic structure of the first innexin-6 gap junction channel of *C. elegans* determined by using single particle cryo-EM. It provided the first glance of this gap junction sub-family, and enabled direct comparison at atomic resolution level with that of the connexin-26. Novel structural insights were visualized for the first time. Technically, structural determination of gap junction has been a major challenge in structural biology, because its double transmembrane domain of the full channel. Not only for X-ray crystallography, this is also a very challenging target for single particle cryo-EM, even with the recent technological advances. Apparently, a number of novel approaches, particularly the sample preparation by using GraDeR to remove detergent micelles played significant role in improving image quality. The figures shown in Supplementary Figure 1 are very intriguing. The atomic structure presented in this manuscript represents a major step forward in the field, both biology and technology. Thus, the scientific significance present in this manuscript is sufficient to merit publishing in Nature Communication.

While the science presented in the manuscript is of excellent quality, the manuscript itself is not perfectly written. If accepted, it may require some editorial efforts to clarify some sentences. I listed a few confusing sentence here, as a minor issue. Otherwise, my specific comments are listed below:

Specific comments:

How was a hemichannel reconstructed? From the boxed particles shown in Supplementary Figure 2a, it seems that the hemichannel dataset contains a mixture of single end of the full channel and individual hemichannel. I assume that full channel and hemichannel cannot be separated by size exclusion chromatography. Particle selection is then a perfect way to separate them computationally. However, it is not clear what is the purpose of reconstructing a hemichannel by using one end of a full channel? Structurally, it would be interesting to see if a hemi-channel has a different conformation from a full channel. But the way used to determine a hemichannel reconstruction would not be able to reveal such difference, if there is any. If the goal is to increase the resolution, why not combine the full channel (D8) with hemichannel (C8), assuming there is no conformation difference between the two?

Minor:

First sentence of page 5: “The map of the INX-6 hemichannel partially presents the density ...”
The sentence reads strange, probably should change to “ The map ... include part of the densities of the ..”?

The sentence “Using the atomic model of the INX-6 hemichannel as a reference, we constructed a model of the INX-6 gap junction channel (Fig. 1e).” is confusing. It seems from Supplementary Figure 2 that the full channel is reconstructed ab initio?

In method section the dose rate should use physical pixel (1.232Å) or state clearly the pixel is super-resolution pixel (0.616Å).

Was binning by Fourier cropping or real space averaging?

The sentence “For example, NT ...” is too long and fragmented. It is better to break it into multiple sentences.

Reviewer #2 (Remarks to the Author):

Oshima et al reported the cryo-electron microscopy structure of the *Caenorhabditis elegans* innexin-6 (INX-6) gap junction channel at atomic resolution. They found the arrangements of the transmembrane helices, extracellular loops, and the N-terminal pore funnel on the INX-6 monomeric structure are similar to those of Cx26, despite the of significant sequence similarity. The helix-rich cytoplasmic loop and C-terminus are intercalated one-by-one through an octameric hemichannel, forming a dome-like entrance that interacts with N-terminal loops in the pore. The data suggest that INX-6 cytoplasmic domains are cooperatively associated with the N-terminal funnel conformation. Overall, the manuscript provides significant information about structure and potential function of gap junction channels. The observation of the dome-like structure formed by the cytoplasmic domains of INX-6 is exciting and will be interesting to see if any of this cooperatively translates to the vertebrate gap junction channels.

Minor

- 1) The authors mention in the introduction that the C-terminal domain of gap junction channels have not observed previously in any connexin structures. This statement is not entirely correct. Rosslyn et al 2012 (Biomol NMR assignments) determined that membrane tethering of the CT to the 4th transmembrane domain was needed for the CT domain to adopt helical structure (albeit not a rigid structure). Based upon the authors study, one may envision that the Cx43CT in context of a full connexin and in a connexon, has the potential to adopt a structure as presented for INX-6. The reviewer feels this should be addressed.
- 2) Can the authors explain why in Fig 1 d (TM1: middle right; TM3: middle right; TM4 upper right) amino acid side chains look outside the electron density? Is there flexibility in these areas of the pore?
- 3) Would be really helpful to provide a vertical cross-section through the gap junction channel, showing the surface potential inside the channel (please see Maeda et al 2009; Figure 4)

Response to reviewers:

We thank both reviewers for their constructive and helpful comments on this manuscript.

Specific answers to Reviewer #1 comments:

Reviewers' comments:

Reviewer #1 (Remarks to the Author):

Gap junction mediates cell-cell communication by letting relatively large substrates pass through from cytosol of one cell to another. It plays critical role in maintaining cellular homeostasis in multicellular organisms. The gap junction in vertebrate is named connexin and in invertebrate is named innexin. The two shares no apparent sequence homology, but its overall structural architectures were predicted to be somewhat similar. The atomic structure of connexin-26 was determined by X-ray crystallography a few years ago. However, no structure information of innexin was available, not even its oligomeric state. This manuscript by Oshima et al described the atomic structure of the first innexin-6 gap junction channel of *C. elegans* determined by using single particle cryo-EM. It provided the first glance of this gap junction sub-family, and enabled direct comparison at atomic resolution level with that of the connexin-26. Novel structural insights were visualized for the first time. Technically, structural determination of gap junction has been a major challenge in structural biology, because its double transmembrane domain of the full channel. Not only for X-ray crystallography, this is also a very challenging target for single particle cryo-EM, even with the recent technological advances. Apparently, a number of novel approaches, particularly the sample preparation by using GraDeR to remove detergent micelles played significant role in improving image quality. The figures shown in Supplementary Figure 1 are very intriguing. The atomic structure presented in this manuscript represents a major step forward in the field, both biology and technology. Thus, the scientific significance present in this manuscript is sufficient to merit publishing in Nature Communication.

While the science presented in the manuscript is of excellent quality, the manuscript itself is not perfectly written. If accepted, it may require some editorial efforts to clarify some sentences. I listed a few confusing sentence here, as a minor issue. Otherwise, my specific comments are listed below:

Specific comments:

How was a hemichannel reconstructed? From the boxed particles shown in Supplementary Figure 2a, it seems that the hemichannel dataset contains a mixture of single end of the full channel and individual hemichannel. I assume that full channel and hemichannel cannot be separated by size exclusion chromatography. Particle selection is then a perfect way to separate them computationally. However, it is not clear what is the purpose of reconstructing a hemichannel by using one end of a full channel? Structurally, it would be interesting to see if a hemi-channel has a different conformation from a full channel. But the way used to determine a hemichannel reconstruction would not be able to reveal such difference, if there is any. If the goal is to increase the resolution, why not combine the full channel (D8) with hemichannel (C8), assuming there is no conformation difference between the two?

Authors:

We agree that the micrographs we used include both docked and undocked hemichannel particles which are of course mixed in the hemichannel particle data set. In 3D classification, the 3D structure of class 1 represents a single end hemichannel of the docked full channel (we call docked hemichannel), and class 3 shows an undocked hemichannel (Supplementary Fig. 2d). This demonstrates that RELION tells docked and undocked hemichannels at least in side view orientations. While class 1 reached 3.3 Å resolution finally, class 3 produced a 7.3 Å resolution map with the angular distribution highly biased toward top views. We further selected only side view particles from class 3 by 2D classification. The 3D structure was however reconstructed from the selected 6401 particles at 6.0 Å resolution. Because the particle number was insufficient for an atomic resolution and it was unclear if this structure of class 3 has a significantly different conformation, we used the structure from class 1 of which the high resolution enabled us to build a reliable *de novo* model.

To clarify this, we added the 3D structures and angular distributions after 3D refinement of class 3 to Supplementary Fig. 2d. Methods and the legend of Supplementary Fig. 2d were revised to note that the class 3 particles represented a 3D map corresponding to an undocked hemichannel, but its resolution did not reach atomic level. We also inserted the experimental flow chart as Supplementary Fig. 5.

p. 5

“The processing schemes we used for the hemichannel and gap junction channel are summarized in a flow chart (Supplementary Fig. 5).”

p.12

“Three-dimensional classification with C1 symmetry produced five classes. “Class 1” which showed the finest features of a docked hemichannel structure derived from 74,398 particles and “class 3” which contained 77,697 particles showing the features of an undocked hemichannel were independently subjected to the next iterative refinement with C8 symmetry. The three-dimensional refinement of “class 3” reached 7.3 Å resolution with the angular distribution biased toward top views (Supplementary Fig. 2d). The subsequent two-dimensional classification allowed us to select only side view particles from “class 3”. However, the selected 6401 particles produced a 6.0 Å resolution map (Supplementary Fig. 2d), which was not included in this work due to the low resolution.”

As for the last question, currently we do not think that the full channel (D8) should be combined with the hemichannel (C8) because we found some differences between these two, if not biologically significant. One is map quality. While the resolution of a full channel map (D8) became close to an atomic resolution (3.6Å), the side chain features are not well resolved as compared with the hemichannel map (C8) (the following figure (a), see also Supplementary Fig. 3a). The other thing is different rotamers of a couple of side chains in E1 (the following figure (b)). While we speculate there is a possibility that the differences are caused by a mixture of docked and undocked particles in top view orientations that would be

indistinguishable from each other, 3D reconstruction using only the side view particles did not improve the map (data not shown). To increase the resolutions of these structures, we would have to figure out a way to prepare pure undocked hemichannels and/or pure docked junction channels, not a mixture of them. At present, we think it is more careful to keep these two separate.

Reviewer #1

Minor:

First sentence of page 5: “The map of the INX-6 hemichannel partially presents the density ...” The sentence reads strange, probably should change to “The map ... include part of the densities of the ..”?

Authors:

We thank the reviewer for pointing this out. This sentence has been fixed according to the suggestion.

p.5

“The map of the INX-6 hemichannel includes part of the densities of the extracellular domains of an opposed hemichannel (Fig. 1b).”

Reviewer #1

The sentence “Using the atomic model of the INX-6 hemichannel as a reference, we constructed a model of the INX-6 gap junction channel (Fig. 1e).” is confusing. It seems from Supplementary Figure 2 that the full channel is reconstructed ab initio?

Authors:

The 3D maps of hemi- (C8) and full (D8) channels were reconstructed independently. However, the model of a full channel was not ab initio, but was generated using the hemichannel coordinates because of insufficient quality of the 3D map for *de novo* modeling. To avoid confusion, the sentence has been revised as follows. We also hope that the flow chart (Supplementary Fig. 5) would be helpful.

p.5

“A model of the INX-6 gap junction channel was refined after fitting the atomic model of the INX-6 hemichannel into the density map of the INX-6 gap junction channel.”

Reviewer #1

In method section the dose rate should use physical pixel (1.232Å) or state clearly the pixel is super-resolution pixel (0.616Å).

Authors:

We revised the dose rate according to the suggestion, and exactly described the pixel size.

p.11

“The dose rate was limited to 10.8 e⁻ per physical pixel per second....”

p. 11

“.....a pixel size of 1.232 Å.”

Reviewer #1

Was binning by Fourier cropping or real space averaging?

Authors:

The images were binned by Fourier cropping. We revised Methods.

p.11

“For image processing, the dose fractioned image stacks were binned 2×2 by Fourier cropping,.....”

Reviewer #1

The sentence “For example, NT ...” is too long and fragmented. It is better to break it into multiple sentences.

Authors:

We appreciate this suggestion. The sentence has been modified as follows.

p. 5

“Specifically, the N-terminus has a short helix (NTH) facing the pore. The

assigned order of transmembrane helices is the same, and the innermost helix is TM1. TM2 is kinked by a proline. E1 contains a small α -helix (E1H), and anti-parallel β -sheets are formed in E2 (Fig. 2a, b and Fig. 3a).”

Specific answers to Reviewer #2 comments:

Reviewer #2 (Remarks to the Author):

Oshima et al reported the cryo-electron microscopy structure of the *Caenorhabditis elegans* innexin-6 (INX-6) gap junction channel at atomic resolution. They found the arrangements of the transmembrane helices, extracellular loops, and the N-terminal pore funnel on the INX-6 monomeric structure are similar to those of Cx26, despite the of significant sequence similarity. The helix-rich cytoplasmic loop and C-terminus are intercalated one-by-one through an octameric hemichannel, forming a dome-like entrance that interacts with N-terminal loops in the pore. The data suggest that INX-6 cytoplasmic domains are cooperatively associated with the N-terminal funnel conformation. Overall, the manuscript provides significant information about structure and potential function of gap junction channels. The observation of the dome-like structure formed by the cytoplasmic domains of INX-6 is exciting and will be interesting to see if any of this cooperatively translates to the vertebrate gap junction channels.

Minor

1) The authors mention in the introduction that the C-terminal domain of gap junction channels have not observed previously in any connexin structures. This statement is not entirely correct. Rosslyn et al 2012 (Biomol NMR assignments) determined that membrane tethering of the CT to the 4th transmembrane domain was needed for the CT domain to adopt helical structure (albeit not a rigid structure). Based upon the authors study, one may envision that the Cx43CT in context of a full connexin and in a connexon, has the potential to adopt a structure as presented for INX-6. The reviewer feels this should be addressed.

Authors:

We agree that our expression in the first version was misleading. Introduction was revised and an additional sentence was added in Discussion citing the reference that the reviewer suggested.

p.3

“The detailed features of the cytoplasmic loop (CL) and C-terminal domain (CT) of gap junction channels have not been visualized in any crystallographic studies on connexins”

p. 9

“It has also been reported that tethering of the CT of Cx43 to TM4 induces the CT domain to adopt helical structure³³, which suggests the potential of similar cytoplasmic arrangements between Cx43 and INX-6. While other connexins.....”

Reviewer #2

2) Can the authors explain why in Fig 1 d (TM1: middle right; TM3: middle right; TM4 upper right) amino acid side chains look outside the electron density? Is there flexibility in these areas of the pore?

Authors:

We apologize for having used a pre-final model in Fig. 1d in the first version. Fig. 1d has been replaced with the final model and the side chain of W204 (TM3: middle right) now fits to the density. For the side chains of TM1 and TM3, we agree with the reviewer that this is probably due to the flexibility of side chains. The lower contour levels such as 1σ and 0.5σ cover those side chains, but make the densities bold and noisy. We used a contour level of 2σ in the main figure because of showing the finest features of most side chains. We described the map contour level (2σ) in the legend of Fig. 1d.

p.20

“The density map is contoured at 2.0σ .”

Reviewer #2

3) Would be really helpful to provide a vertical cross-section through the gap junction channel, showing the surface potential inside the channel (please see Maeda et al 2009;

Figure 4)

Authors:

Supplementary Fig. 8a has been modified to clearly show the surface potential inside the pore pathway with increased slab width. The N-terminal funnel and extracellular constrictions were annotated in the figure. We hope this would be more comprehensible.

We also cited Supplementary Fig. 8 in Discussion.

p.8

“Although a positively charged environment at the Cx26 channel entrance may possibly contribute to the charge selectivity of permeates⁹, the positive pore pathway of INX-6 (Supplementary Fig. 8a) would be less related to the selectivity because of large diameter.”

Reviewers' Comments:

Reviewer #1 (Remarks to the Author):

No further comments.

Reviewer #2 (Remarks to the Author):

The authors addressed all of the reviewers concerns.